# Maternal Blood Concentration of Tadalafil and Uterine Blood Flow in Pregnancy

**DOI:** 10.3390/medicina55100708

**Published:** 2019-10-21

**Authors:** Hiroaki Tanaka, Shintaro Maki, Shoichi Magawa, Masafumi Nii, Kayo Tanaka, Kenji Ikemura, Kuniaki Toriyabe, Tomoaki Ikeda

**Affiliations:** 1Department of Obstetrics and Gynecology, Mie University School of Medicine, 2-174 Edobashi, Tsu, Mie 514-8507, Japan; mabochikin519@yahoo.co.jp (S.M.); shoichimagawa@yahoo.co.jp (S.M.); doaldosleeping@yahoo.co.jp (M.N.); tagami.t.ky@gmail.com (K.T.); to.kuniaki@gmail.com (K.T.); t-ikeda@clin.medic.mie-u.ac.jp (T.I.); 2Department of Clinical Pharmacy and Biopharmaceutics, Mie University Hospital, 2-174 Edobashi, Tsu, Mie 514-8507, Japan; ikemurak@clin.medic.mie-u.ac.jp

**Keywords:** tadalafil, pregnancy, preeclampsia, fetal growth restriction

## Abstract

*Background and Objectives:* Tadalafil for treatment of fetal growth restriction (FGR) or preeclampsia is given once a day orally. The drug kinetics of tadalafil were investigated to determine the ideal dosage to promote uterine blood flow. *Materials and Methods:* We recruited five pregnant women with FGR or preeclampsia before administration of tadalafil, all of which were administered tadalafil (20 mg/day, once-daily dosing). The blood concentration of tadalafil was measured 1, 2, 4, 6, 8, and 24 h after administration, and uterine blood flow was measured before tadalafil administration and 2–4 and 20–24 h after. We then analyzed the correlation between tadalafil blood concentration and uterine artery blood flow. *Results:* The blood concentration of tadalafil correlated with uterine artery blood flow in pregnant women. The blood concentration of tadalafil and uterine artery blood flow decreased 5 h after administration of tadalafil. *Conclusions:* The blood concentration of tadalafil and uterine artery blood flow fluctuate in parallel, the latter was decreased by reduced blood concentration. Thus, a study of tadalafil administered twice a day in pregnant women will be needed to stabilize uterine artery blood flow.

## 1. Introduction

Fetal growth restriction (FGR) is a common complication of pregnancy that is associated with various adverse perinatal outcomes [1]. However, there is currently no proven fetal therapy to reverse or ameliorate overt FGR. To prevent FGR, nutritional and dietary supplementation, bed rest, and aspirin therapy have been investigated, but the efficacy of these treatments is not supported by sufficient evidence [2].

Therefore, development of a novel therapy for FGR has been a focus in many countries. Tadalafil, a phosphodiesterase 5 (PDE5) inhibitor, was reported to be one of these novel therapies for FGR [3,4,5,6,7]. Although several other PDE5 inhibitors are available, such as sildenafil, tadalafil has a longer half-life than sildenafil (14–15 h vs. 2–4 h) and is consequently presumed to be more stable and effective [8]. Another benefit of the longer half-life of tadalafil is that one dose per day is sufficient, whereas sildenafil must be administered at least twice per day. Although PDE5 enzymes are widely distributed in blood vessels, tadalafil is particularly selective for PDE5 enzymes found in the reproductive organs [9]. In fact, the TADAFER II study revealed that tadalafil administration prolonged gestation, in contrast to the STRIDER study [10]. Sildenafil, as a PDE5 inhibitor, did not prolong gestation in FGR in the STRIDER study [11,12]. The differences between the STRIDER study and the TADAFER study are shown in Table 1.

For the effective use of tadalafil for FGR, the aim of this study was to identify the relationship between the serum concentration of tadalafil and uterine artery blood flow in pregnant women and determine the ideal dosage to promote uterine artery blood flow.

## 2. Materials and Methods

### 2.1. Subjects

We recruited all pregnant women who were receiving tadalafil treatment for FGR from April 2016 to November 2018. All women who gotten informed consent were registered in this study. It was conducted at a single institution in Japan. The dose of tadalafil was 20 mg per day in all cases (oral administration). The dose of tadalafil was determined by our phase I study for FGR [5]. FGR was defined as 1.5 standard deviations below the mean estimated fetal body weight according to gestation as calculated from ultrasonography. We received informed consent from all study subjects according to the Declaration of Helsinki. This trial was approved by the Mie University Faculty of Medicine Ethics Committee (approval number 3121, approval date 10 April 2017).

### 2.2. Maternal Background, Adverse Events Due to Tadalafil, and Serum Tadalafil Concentration

The clinical parameters of each patient, including age, height, weight, systolic blood pressure before tadalafil administration, systolic blood pressure after tadalafil administration, past history, gestational age at tadalafil administration and adverse events due to tadalafil, were considered in the analysis.

The blood concentration of tadalafil in each patient was measured 1, 2, 4, 6, 8, and 24 h after tadalafil administration. Uterine artery blood flow was measured before tadalafil administration and 2–4 h and 20–24 h after tadalafil administration. We then analyzed the correlation between the blood concentration of tadalafil and uterine artery blood flow.

### 2.3. Determination of Serum Tadalafil Concentration

The concentration of serum tadalafil was determined by high-performance liquid chromatography (HPLC) according to a previously described method with slight modifications [13]. Briefly, a 100-L serum sample was alkalized with 400 L of 0.1 M glycine buffer (saturated with NaCl, pH 10.6) and eluted with 1 mL 1-chlorobutane and dichloromethane (4:1, v/v) for 1 h. The organic phase (800 L) was evaporated to dryness, and the residue was dissolved in 50% acetonitrile and subjected to HPLC. The unbound fractions of tadalafil in serum were determined by ultrafiltration using Amicon^®^ Ultra-0.5 mL centrifugal filters (Millipore, Billerica, MA, USA) at 14,000× *g* for 10 min.

HPLC analyses were performed using an HPLC system (LC-20AD, Shimadzu, Kyoto, Japan) connected to a TSKgel^®^ ODS-80Tm 5-m column (150 × 4.6 mm i.d.; Tosho, Tokyo, Japan) and a fluorescence detector (RF-20Axs, Shimadzu). The column temperature was set at 40 °C. The elution of tadalafil was conducted using 0.1% trifluoroacetic acid and acetonitrile (60:40, v/v) at 1 mL/min. The fluorescence detector was operated at excitation and emission wavelengths of 275 nm and 335 nm, respectively. The mean intra- and inter-day precision and accuracy of this assay were estimated by analyzing five replicates at serum tadalafil concentrations of 25, 50, 500, and 750 ng/mL and were determined to be 5.5% and −3.1%, respectively. The limit of detection of tadalafil was 3 ng/mL.

### 2.4. Measurement of Uterine Artery Blood Flow

Before measurements were performed, the pregnant women rested for more than 20 min in the semi-recumbent position. It was confirmed that there were no abnormalities in the measured uterine artery waveform. The appropriate waveform was defined by the absence of the following: an increased systolic/diastolic (A/B) ratio (>2.6) with or without a notch on the uterine artery blood flow velocity waveform or a normal A/B ratio (<2.6) with either a unilateral notch or a bilateral notch on the uterine artery blood flow velocity waveform. The method of visualization of the uterine artery conformed to that of Bower [14]: by placing the transducer in the lower lateral quadrant of the uterus and angling it medially, we identified crossover of the external iliac artery and placed the range gate over the entire diameter of the uterine artery 1 cm distal to that site. The Doppler waveform was measured during three to six heartbeat cycles, and the median of three measurements was used as the measured value. Using a computer, the left and right uterine arterial blood flow rates were measured based on the blood flow velocity, mean time-averaged flow velocity (TAV), and inner diameter of the uterine artery. Standard changes in uterine arterial blood flow that occurred with each gestational week, as reported by Konje et al. [15], were used to evaluate the uterine artery.

The uterine artery Doppler mode of the GE Voluson E8 Expert machine (Chicago, IL, USA) was selected for measurement of the uterine arteries, ensuring maximum standardization of settings. Pulse repetition frequency was adjusted for each examination to ensure the best fit of the waveforms. The following preset variables were used: harmonic setting: mid, power: 100%, gain: −3, C7 M5 P3 E3, SRI II: 2, frequency: low, quality: normal, pulse Doppler wall motion filter: 60 Hz, and sample size: 2 mm. The angle correction was measured within 60 degrees.

### 2.5. Statistical Analysis

Statistical analyses were performed using univariate chi-squared and Mann–Whitney U-tests. *p* values of <0.05 were considered statistically significant. All analyses were performed using the statistical software package, SAS (version 9.4).

## 3. Results

We recruited a total of five pregnant women. The age, height, weight, and systolic blood pressure are shown in Table 2. All women had normal renal function (the range of serum creatinine was 0.32–0.45) and none had preeclampsia at registration.

The kinetics are shown in Table 3.

The blood concentration of tadalafil correlated with uterine artery blood flow, and the blood concentration of tadalafil and uterine artery blood flow decreased from 5 h after administration of tadalafil (Figure 1).

The cumulative frequency of adverse events from tadalafil administration was four women with headache (Grade 1; three women, Grade 2; one woman), and face flush in one woman (Grade 1). There were no severe fetal and neonatal adverse events that seemed to be related to tadalafil. Adverse events were expressed by grade based on the Common Terminology Criteria for Adverse Events v4.0, as with the TADAFER II study [11].

## 4. Discussion

This study investigated the blood concentration and efficacy (uterine artery blood flow) of tadalafil on a single day, with the aim of evaluating the suitability of the current once-daily doses. Once-daily administration poses challenges in maintaining blood drug levels, and efficacy (evaluated in terms of uterine artery blood flow) was not sufficiently maintained. These results suggest that methods for administering tadalafil need to be investigated further.

Previous studies indicated that tadalafil may be effective for treating FGR. Nonetheless, better and more effective administration methods must be considered. Animal experiments that involved continuous administration of tadalafil mixed with water resulted in stable blood drug concentration and efficacy. Regarding human administration, the package insert stipulates a once-daily dosage given the half-life of tadalafil. However, the question of whether this method actually yields a stable tadalafil blood concentration and efficacy remains to be confirmed. We selected uterine artery blood flow as an indicator of efficacy after finding that the blood concentration of tadalafil had decreased 5 hr after administration, while uterine artery blood flow had diminished within the same time frame. These findings suggest that the method for administering tadalafil in humans must be investigated further. We believe that twice-daily administration is superior for yielding a stable blood drug concentration, and we intend to investigate this in the future. As in the present study, we will also investigate efficacy parameters on the basis of uterine artery blood flow.

This study was limited by the small number of participants. In addition, we were unable to measure uterine artery blood flow 6 to 24 h after tadalafil administration. Future studies are advised to measure uterine artery blood flow in more detail.

In summary, in the current once-daily administration of tadalafil, adequate blood concentrations could not be maintained and stable efficacy was not obtained. Therefore, twice-daily or other administration frequencies must be investigated.

## 5. Conclusions

The blood concentration and uterine artery blood flow fluctuate in parallel, the latter was decreased by reduced blood concentration. Thus, a study of tadalafil administered twice a day in pregnant women will be needed to stabilize uterine artery blood flow.

## Figures and Tables

**Figure 1 medicina-55-00708-f001:**
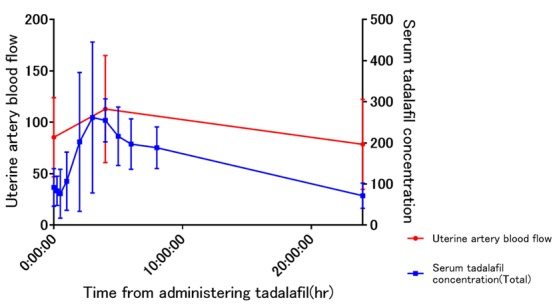
Average concentration–time profiles of tadalafil and uterine artery blood flow. Error bars represent SDs.

**Table 1 medicina-55-00708-t001:** The differences between the STRIDER study and the TADAFER study.

	TADAFER II Study	STRIDER Study
Inclusion criteria	20–34 gestational weeks	22–30 gestational weeks
	1.5 standard deviations below the mean estimated fetal body weight according to gestational as calculated from ultrasonography	Abdominal circumference or estimated fetal weight below the 10 percentile and absent or reversed enddiastolic flow in the umbilical artery on Doppler velocimetry.
The median of prolongation periods (gestational weeks)	Treatment group 47 days Control group 37 days	Treatment group 18 days Control group 18 days

**Table 2 medicina-55-00708-t002:** Clinical parameters of study participants.

Case	Age (Years)	Height (cm)	Weight (kg)	Gestational Age (Weeks)	Systolic BP Before Administration (mmHg)	Systolic BP After Administration (mmHg)	Obstetric Complication	Maternal Complication
1	27	159	56.3	32	110	108	FGR	-
2	33	160	60.9	30	108	105	FGR	Epilepsy
3	22	161	54.3	33	139	145	Preeclampsia	-
4	29	156	51.6	32	98	104	FGR	-
5	35	153.6	59.3	33	108	107	FGR	-

BP, blood pressure; FGR, fetal growth restriction.

**Table 3 medicina-55-00708-t003:** Estimated tadalafil pharmacokinetic parameters in pregnant women.

Parameter	n = 5
AUC_0→24_ (ng·h/mL)	3638 ± 910
AUC_0→24/unbound_ (ng·h/mL)	438 ± 65
CL/F (L/min)	4.13 ± 0.96
V_dss_/F (L)	77 ± 29
C_max_ (ng/mL)	329 ± 134
T_max_ (h)	4 ± 1
Half-life (hr)	13 ± 5

Values are mean ± SD. AUC0→24, area under the tadalafil concentration–time curve from time zero to infinity; CL/F, apparent oral tadalafil clearance; Vdss/F, apparent volume of distribution; Cmax, maximum tadalafil concentration; Tmax, time to maximum concentration.

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
