# Peer review of "Maternal Blood Concentration of Tadalafil and Uterine Blood Flow in Pregnancy"

_medicina, 2019, doi:10.3390/medicina55100708_

Round 1

Reviewer 1 Report

Dear Authors,

PDE5 inhibitors in PE and FGR are promising and require more study, especially after mixed results and safety issues in STRIDER trials. Therefore, your work is valuable, despite the limitation of sample size, since it may help plan a better treatment regime. However, this paper requires extensive editing of English style and language. Below are some suggestions. These are not exhaustive, because peer-review does not imply suggesting a full list of style edits. Keep in mind, that some of problematic phrases occur multiple times in the text.

Abstract

10-11 this sentence is not clear — "tadalafil for treated..." implies, that the condition was treated already. I assume you meant "for treatment of...", but there are other ways to state it.

13 ''were investigated" — don't hesitate to edit out the passive voice and say simply "we investigated the drug kinetics...", like you do in next sentence

14 — it is not clear from the description, whether you recruited already treated women or treated them after recruitment. This may confuse the reader.

17-18 — correlation and coincidence are not interchangeable

20 — do not omit tadalafil when you mention the blood concentration of tadalafil

Keywords

24 — I suggest excluding blood concentration from the keywords. It adds nothing because it is not specific. You may put preeclampsia instead. When revising keywords, imagine the search queries that other research may use to find your research

Introduction

35 - could use a reference for the data on half-life

37 - you may simply refer to sildenafil citrate as sildenafil

39-40 - you may decribe the results of STRIDER and TADAFER II in separate paragraphs, so the reader understand the context of your study. Also, please reference the published results of TADAFER II.

42 — "For increasing the impact of tadalafil for FGR" doesn't really add anything to this sentence

M&M

48-49 — you recruited pregnant and non-pregnant women, but you only report 5 pregnant women in results

51 — please include the Ethics committee protocol number and date

52 — in the subtitle you mention complications (adverse events?), and the say nothing about them in the paragraph

78 — it may be better to change "mother" for "pregnant woman"

Results

104-105 — you don't have to put the data from the table in the text

109-112 — you can simply say, that kinetics are in the Table 2

115-116 — please revise these lines carefully, especially 116. It is not clear what "fu, percent unbound" stands for

123-124 — please report AEs in more details and use Grades if possible. Refer to TADAFER II paper for a how-to

Author Response

Dear Dr. Editors,

We greatly appreciate the reviewers’ constructive comments. According to these comments, we have revised our manuscript. We hope that the revised version meets all the requirements and is now acceptable for publication. Page and line numbers listed in this letter are those of the revised manuscript unless otherwise noted.

Best regards,

Hiroaki Tanaka, MD, PhD

Department of Obstetrics and Gynecology, Faculty of Medicine, Mie University

2-174 Edobashi, Tsu, Mie, Japan

Telephone: +81-059-232-1111

Fax: +81-059-231-5202

Reviewer 1

Dear Authors,

PDE5 inhibitors in PE and FGR are promising and require more study, especially after mixed results and safety issues in STRIDER trials. Therefore, your work is valuable, despite the limitation of sample size, since it may help plan a better treatment regime. However, this paper requires extensive editing of English style and language. Below are some suggestions. These are not exhaustive, because peer-review does not imply suggesting a full list of style edits. Keep in mind, that some of problematic phrases occur multiple times in the text.

 Abstract

10-11 this sentence is not clear — "tadalafil for treated..." implies, that the condition was treated already. I assume you meant "for treatment of...", but there are other ways to state it.

I have revised the sentence as follow, ‘Tadalafil for treatment of fetal growth restriction (FGR) or preeclampsia is given once a day orally in a basic way.’

 13 ''were investigated" — don't hesitate to edit out the passive voice and say simply "we investigated the drug kinetics...", like you do in next sentence

I have revised the sentence as follow, ‘The drug kinetics of tadalafil investigated to determine the ideal dosage to promote uterine blood flow.’

 14 — it is not clear from the description, whether you recruited already treated women or treated them after recruitment. This may confuse the reader.

I have revised the sentence as follow, ‘We recruited five pregnant women with FGR or preeclampsia before administration of tadalafil.’

 17-18 — correlation and coincidence are not interchangeable

I have revised the sentence from coincidence to correlation.

20 — do not omit tadalafil when you mention the blood concentration of tadalafil

I have revised the sentence from blood concentration to blood concentration of tadalafil

 Keywords

24 — I suggest excluding blood concentration from the keywords. It adds nothing because it is not specific. You may put preeclampsia instead. When revising keywords, imagine the search queries that other research may use to find your research

I have excluded ‘blood concentration’ and added ‘preeclampsia’ in keywprds.

 Introduction

35 - could use a reference for the data on half-life

We have added a reference.

 37 - you may simply refer to sildenafil citrate as sildenafil

I have delated ‘citrate’.

 39-40 - you may describe the results of STRIDER and TADAFER II in separate paragraphs, so the reader understand the context of your study. Also, please reference the published results of TADAFER II.

I have added the new Table showed the result in TRIDER and TADAFER II. And I added the reference.

 42 — "For increasing the impact of tadalafil for FGR" doesn't really add anything to this sentence

I have revised the sentence as follow, ‘For the effective use of tadalafil for FGR’.

 M&M

48-49 — you recruited pregnant and non-pregnant women, but you only report 5 pregnant women in results

I have delated the sentence as follow, ‘We also recruited non-pregnant women, who were likewise administered tadalafil (20 mg/day) in the same periods.’.

 51 — please include the Ethics committee protocol number and date

I have added the ethics committee protocol number and date.

 52 — in the subtitle you mention complications (adverse events?), and the say nothing about them in the paragraph

I have added the adverse events due to tadalafil as follow.

Cumulative frequency of adverse events from tadalafil administration were 4 women in headache (Grade 1; 3 women, Grade 2; 1 woman), face flash in 1 woman (Grade 1). There were no severe fetal and neonatal adverse events that seemed to be related to tadalafil. Adverse events were expressed by grade based on the Common Terminology Criteria for Adverse Events v4.0 as with TADAFERâ…¡study ï¼»11ï¼½.

 78 — it may be better to change "mother" for "pregnant woman"

I have revised the sentence as suggested by reviewer.

 Results

104-105 — you don't have to put the data from the table in the text

I have delated the data from the table.

 109-112 — you can simply say, that kinetics are in the Table 2

I have revised the simply say.

 115-116 — please revise these lines carefully, especially 116. It is not clear what "fu, percent unbound" stands for

I have revised the sentence as suggested by reviewer.

 123-124 — please report AEs in more details and use Grades if possible. Refer to TADAFER II paper for a how-to

I have revised the sentence as suggested by reviewer, and added reference as follow.

Cumulative frequency of complication from tadalafil administration were 4 women in headache (Grade 1; 3 women, Grade 2; 1 woman), face flash in 1 woman (Grade 1). There were no severe fetal and neonatal complications that seemed to be related to tadalafil. Complication were expressed by grade based on the Common Terminology Criteria for Adverse Events v4.0 as with TADAFERâ…¡study ï¼»15ï¼½.

Reviewer 2 Report

This is well written interesting manuscript evaluating the maternal concentration of tadalafil in pregnancies c/w IUGR to determine the frequency of daily use. Authors used small group of participants to evaluate blood concentrations of tadalafil and it is association with uterine artery blood flow. 

Scientifically, this study study sounds interesting to the Medicina readers, however, number of cases are very small and I have several recommendations to the authors for this study becomes eligible for publication. 

1) Materials and methodology is concise, needs more descriptive analysis. How was sample collection performed? what is the SOP for collection? 

2) What is the percentile of IUGR cases. 

3) Tadalafil is eliminated primarily as metabolites in faeces (61%) and urine (36%). Please discuss if there was a confounding factor of diminished renal function, especially with included preeclampsia case. 

4) Please discuss the reason of including non-pregnant women? Also what was the number of subjects who were not pregnant?

Minor comments:

1) Authors have inconsistently used uterine blood flow and uterine artery blood flow. I recommend having consistent statement of uterine artery blood flow. 

2) Please state IRB number with corresponding date. 

Author Response

Dear Dr. Editors,

We greatly appreciate the reviewers’ constructive comments. According to these comments, we have revised our manuscript. We hope that the revised version meets all the requirements and is now acceptable for publication. Page and line numbers listed in this letter are those of the revised manuscript unless otherwise noted.

Best regards,

Hiroaki Tanaka, MD, PhD

Department of Obstetrics and Gynecology, Faculty of Medicine, Mie University

2-174 Edobashi, Tsu, Mie, Japan

Telephone: +81-059-232-1111

Fax: +81-059-231-5202

Reviewer 2

 This is well written interesting manuscript evaluating the maternal concentration of tadalafil in pregnancies c/w IUGR to determine the frequency of daily use. Authors used small group of participants to evaluate blood concentrations of tadalafil and it is association with uterine artery blood flow.

Scientifically, this study study sounds interesting to the Medicina readers, however, number of cases are very small and I have several recommendations to the authors for this study becomes eligible for publication.

1) Materials and methodology is concise, needs more descriptive analysis. How was sample collection performed? what is the SOP for collection?

Thank you for your comment. We have received the ‘subject’ in methods section. We recruited all pregnant women receiving tadalafil treatment for FGR from April 2016 to November 2018. The all pregnant women obtained informed consent were registered in this study. It was conducted at a single institution. The dose of tadalafil was 20 mg per day in all cases (oral administration). The dose of tadalafil determined by our phase I study for FGRï¼»5ï¼½. We received informed consent from all study subjects according to the Declaration of Helsinki. This trial was approved by the Mie University Faculty of Medicine ethics committee (approval number; 3121, corresponding date; April, 10th, 2017).

 2) What is the percentile of IUGR cases.

All women had fetal growth restriction (FGR).

I have added the percentage of FGR in result section.

3) Tadalafil is eliminated primarily as metabolites in faeces (61%) and urine (36%). Please discuss if there was a confounding factor of diminished renal function, especially with included preeclampsia case.

All women were normal renal function (The range of serum creatine 0.32-0.45) and there is no woman with preeclampsia in this study.

I had added the information of maternal renal function.

4) Please discuss the reason of including non-pregnant women? Also what was the number of subjects who were not pregnant?

I have delated the word of ‘non-pregnant’. The word of ‘Non-pregnant’ is miss sentence.

 Minor comments:

 Authors have inconsistently used uterine blood flow and uterine artery blood flow. I recommend having consistent statement of uterine artery blood flow. I have revised to uterine artery blood flow.

 2) Please state IRB number with corresponding date.

I have added IRB number with corresponding date.

Round 2

Reviewer 2 Report

The authors have incorporated most of the recommendations suggested in the main text. Their point to point reply to both reviewers is clear and convincing. However, I have several minor concerns:

1)  Authors stated that "All women werenormal renal function(The range of serum creatine 0.32-0.45)and there is no woman with preeclampsia in this study.", however, Table 2 has a patient diagnosed with preeclampsia with mild range Systolic blood pressure. So statement does not match with Table information. 

2) Authors stated that they've included percentile of FGR cases. Please indicate where they are presented, also I recommend adding them to Table 2. Also, in methodology, what percentile scale is used for diagnosis, local population based vs national percentile chart? 

Author Response

The authors have incorporated most of the recommendations suggested in the main text. Their point to point reply to both reviewers is clear and convincing. However, I have several minor concerns:

Authors stated that "All women werenormal renal function(The range of serum creatine 0.32-0.45)and there is no woman with preeclampsia in this study.", however, Table 2 has a patient diagnosed with preeclampsia with mild range Systolic blood pressure. So statement does not match with Table information. 

 Thank you for your comment. We have revised from in this study to at registration.

All women were normal renal function (The range of serum creatine 0.32-0.45) and there is no woman with preeclampsia at registration.

Authors stated that they've included percentile of FGR cases. Please indicate where they are presented, also I recommend adding them to Table 2. Also, in methodology, what percentile scale is used for diagnosis, local population based vs national percentile chart? 

We have added the definition of FGR

FGR was defined 1.5 standard deviations below the mean estimated fetal body weight according to gestational as calculated from ultrasonography in Japan.